# Research on Sustainable Economic Dynamics: Digital Technology Development and Relative Poverty of Urban Households

Sensen Jin  and Feng Deng *

School of Economics and Management, Xinjiang University, Urumqi 830046, China; jss1207@163.com
* Correspondence: dengfeng@xju.edu.cn

**Abstract:** The digital application gap and relative poverty caused by the development of digital technology are both important factors affecting sustainable economic dynamics. This paper explores the impact of digital technology development on the relative poverty of urban households in China, using the China Household Tracking Survey CFPS2010–2018 and the word frequency crawling technology of Python software. The results indicate that failure to adapt to the demands of digital technology may result in a multidimensional digital technology application gap, leading to increased income inequality among urban households and a higher likelihood of relative poverty. In economically developed areas, households headed by individuals with low levels of education and high levels of family support should be particularly mindful of the phenomenon of digital poverty. This paper expands the research scope of relative poverty and deepens the understanding of the relationship between digital technology and economic development, which is beneficial for the government to accelerate the construction of internal and external support mechanisms and to effectively address the challenges posed by sustainable economic development.

**Keywords:** digital technology development; relative poverty of urban households; digital technology application gap; sustainable economic dynamics



## 1. Introduction

After building a moderately prosperous society in all respects and eliminating absolute poverty, China has embarked on the path of relative poverty alleviation. The alleviation of relative poverty is conducive to raising the level of human capital and ensuring the rational use of resources, and it is an important issue for sustainable economic dynamics. At present, a large amount of literature focuses on the issue of relative poverty in rural areas, and there are studies that have found the important role played by the increasing digitalization of the population in rural poverty reduction. However, with the spread of interconnected sharing of production and convenience of consumption in the digital economy [1], the governance of poverty in China in the new era is no longer limited to rural areas, and the governance of relative poverty needs to focus on both urban and rural areas [2].

The pressure to survive and the struggles of the less fortunate can lead to social unrest and conflict, which can undermine social harmony and have a negative impact on regional economic growth. Scholars have long been concerned with the issue of relative poverty in urban and rural areas. Research has shown that entrepreneurship policies play a crucial role in reducing poverty among low-income urban populations [3]. Additionally, the inadequacy of low-income insurance policies exacerbates urban poverty [4], and the urban and rural dichotomy further contributes to this issue [5]. Precision poverty alleviation have been identified as an effective means of reducing poverty and narrowing the income gap between urban and rural households [6]. Overall, fewer studies have focused on the impact of the digital divide created by the development of digital technology on the relative poverty of urban households.

In recent years, scholars have conducted several studies on the issue of the urban digital divide. Scholars shifted the focus of the digital divide from rural and urban to intra-city and found that the problem of the digital divide within the city cannot be ignored due to geographic and economic limitations [7]. Secondly, regarding the impacts of the digital divide, a study on unemployed workers in Singapore found that the use of information technology is a crucial factor affecting the employment of urban residents [8]. Similarly, a survey conducted on low-income individuals in urban communities in the United States revealed that disconnection among the urban poor is significant and can directly impact information access and employment [9]. Current research on relative poverty in China's towns and cities, such as the study of relative poverty among indigenous and migrant households, has not yet addressed the impact of digital technology adoption on poverty rates. The study found that most of the households that fell into poverty after 2018 were indigenous households in towns and cities [10]. It is important to consider the role of digital technology adoption in future research on poverty in China.

Digital technologies, such as Big Data, cloud computing, blockchain, artificial intelligence and 5G, are causing a digital transformation in various industries. This transformation is characterized by changes in production processes. In recent years, digital technologies have rapidly developed and can be integrated into numerous industries [11], becoming a key factor when influencing sustainable economic dynamics. They are also considered key core technologies, and subsequent empirical evidence uses the proportion of word frequency of digital technologies in government reports and patents for digital technology inventions to represent digital technology development. Digital technology development has created a new form of poverty-causing factor known as the digital skills divide. Acquiring digital skills can act as a catalyst for individuals to escape poverty [12]. Regarding the division of individual skills, individuals who work with computers generally possess higher digital skills compared to those who only use mobile phones to access the Internet [13]. This is because computers offer the necessary space to read, write and create complex content, whereas the mobile phone's Internet interface has limited access to important content [14]. This paper's micro-study of the digital skills divide focuses on the acquisition of basic ICT skills, specifically access to computers and mobile phones.

This paper's potential contributions are two-fold. Firstly, it includes both government and enterprise levels in its portrayal of digital technology development. This is achieved by using the frequency of key digital technology words in the government's work report to represent the intensity of digital technology development, and the number of digital technology invention patents of enterprises to represent the breadth of digital technology development. Secondly, this paper conducts a robustness test mainly using the latter measure. The second contribution serves to verify the positive relationship between the development of digital technology and urban relative poverty from the micro household perspective. This complements the inadequacy of existing studies that focus solely on the relative poverty of rural households. The third contribution is the construction of a systematic framework for analyzing the digital divide. This framework specifically measures the horizontal digital divide between industries and the vertical digital divide formed between the government and households. The aim is to explain how digital technology development externally affects the relative poverty of urban households. The fourth contribution is to explore the internal mechanisms by which the development of digital technology affects the relative poverty of urban households, from the perspective of income inequality among households.

This study is structured as follows: Section 2 reviews the existing research literature to identify the impacts and mechanisms of digital technology development on the relative poverty of urban households. Section 3 presents the model construction and indicator selection. Section 4 presents the results of the empirical analyses, including a discussion of robustness and heterogeneity. Section 5 analyzes the impact mechanism, which is divided into two dimensions: the divide in digital technology application and inter-household income inequality. Section 6 presents the study's conclusions and implications.

## 2. Literature Review and Research Hypothesis

### 2.1. Relative Poverty Measurement and Relative Poverty of Urban Households

Relative poverty refers to the deprivation of economic rights and social welfare in comparison to overall social development. Academic research on relative poverty currently focuses on its measurement and identification, which can be categorized into two aspects: strong relative poverty and weak relative poverty. Fuchs (1967) argued that relative income could be represented by 50 percent of the median per capita income, a view shared by Eurostat and OECD countries [15]. Some scholars have expanded the median to 60 percent [16], while the others used 50 percent or 60 percent of the mean income to measure relative poverty [17]. Such measures enable a relatively fixed elasticity of the poverty line concerning the overall income level. They provide a more accurate representation of income disparities and are supported by the monetary utility theory and the theory of differential costs of social inclusion. Therefore, they are referred to as "strong relative poverty lines".

However, the overall income benefit from increasing economic growth should be considered in relation to the relative ability deprivation. The relative poverty line should decline accordingly. It is important to adopt a strong relative poverty line that reflects income level increases and gradually becomes higher. This will help to identify individual poverty. However, it is important to avoid the excessive identification of relative poverty. Subsequently, it is important to add social inclusion lower-limit parameters to the perspective of individual survival and social inclusion, including absolute poverty and a relative poverty line [18]. This formed a more widely applicable "weak relative poverty line". They further used the discounted Gini mean to measure income, gradually correcting and enriching the weak relative poverty measure and identification theory system. Jolliffe and Prydz developed the SPL index method by combining the relative poverty measure method with the international absolute poverty standard of USD 1.20 per person per day (converted to USD 1.90 in terms of purchasing power) [19]. Moreover, the minimum living expenditure in the SPL index was used to calculate the subjective relative poverty line and assess its validity [20]. In summary, the research conducted thus far has established a comprehensive theoretical framework, providing a solid foundation for further empirical studies.

Early research on urban family poverty in China has primarily focused on the stages of China's economic transition reform and the expansion of urban income inequality [21]. Additionally, research has been conducted on the rising inequality, food prices, and income uncertainty contributing to urban poverty [22]. Scholars focus on the changes and causes of absolute poverty in urban China, including the importance of entrepreneurship policy in poverty reduction among urban low-income people and the alleviation and deficiency of urban poverty caused by the subsistence allowance policy [3,4,23–25]. In 2007, the urban–rural dual pattern intensified urban poverty [5]. Targeted poverty alleviation programs play an important role in reducing poverty and narrowing the income gap between urban and rural families.

The study on relative poverty in urban development in China began relatively recently and was limited in scope. The social poverty line (SPL) index method was used to establish the relative poverty standard. It was found that the issue of relative poverty was more prevalent in rural areas than in urban areas [26]. In their analysis of the 2013 and 2018 Chinese Family Tracking Survey data (CFPS), it was found that the median per capita income and wealth of urban families and migrant families were fixed in proportion. They also discovered that the gap in relative poverty between urban immigrant families and native families narrowed over time [10]. The study highlights the changing and complex nature of urban relative poverty, which requires further attention. Therefore, studying the changes in the income structure of urban households and their impact on the occurrence of relative poverty is of great practical significance in the context of China's rapid urbanization process and the complexity of urban poverty. This study will focus on the progress of digital technology.

### 2.2. The Influence of Digital Technology Development on Urban Poverty

For a considerable period, the academic community has focused on responding to urban relative poverty. Existing research has acknowledged the positive impact of digital economy development on urban poverty reduction. Eleftheriadou et al. [27] argue that cutting-edge technologies can drive a digital resurgence to address global urban problems such as poverty and inequality. Das and Chatterjee investigate the impact of information and communication technology (ICT) on poverty reduction in India through digital financial inclusion. However, the proliferation of ICT technology in the banking sector weakens this effect [28].

Digital technology is a key driver of the digital economy, and its frequent appearance in Chinese local government work reports highlights its importance. The government's increased focus on digital technology will inevitably lead to its wider adoption across various industries, promoting the digital transformation of the industry. The gradual integration of digital technology into various industries of the national economy may result in an increase in the use of the industrial Internet, industrial robots and other applications. This could potentially reduce the employment opportunities for urban laborers, leading to a decline in employment stability and creating a risk of relative poverty. Differences in the government's approach to developing various digital technologies can result in disparities in their diffusion, leading to regional and sectoral imbalances in the digital economy's development. This can cause unequal benefits for urban households and potentially increase relative poverty among them.

**Hypothesis 1.** *Digital technology development may increase the incidence of relative poverty among urban households.*

### 2.3. The Digital Technology Adoption Divide and Relative Urban Poverty

The digital skills application divide is gradually widening due to the increasing penetration of digital development orientation and the spread of Internet access at the physical level [29]. This phenomenon is often linked to the issue of urban poverty. In the application of urban digital technology, existing research has highlighted the gap in its application. Reddick, Enriquz and Sharma [7] found that residents in low-income areas have a very low broadband utilization rate, indicating a clear digital divide within cities. Caragliu and Del's study found that the uneven diffusion of smart city technology may exacerbate the differentiation in human resources, leading to income inequality in cities [30]. Tanbenbock et al. used Twitter geographic location data to study the relationship between urban economic differences and the digital divide [31]. They found that digital backwardness exists in urban slum areas, and fair access to geographic information technology for the edge of community residents is difficult. Therefore, using open geographic information system software to fight poverty is a good choice [32]. It can be seen that the urban digital technology adoption divide plays a transmission role in the government's digital technology development, affecting relative poverty in towns and cities.

From an industry perspective, there are differences in the ability of firms with varying levels of productivity to absorb and utilize the Internet for disseminating new knowledge. This generates a phenomenon known as the digital divide between industries. Furthermore, the digital transformation of an industry can increase the investment motivation and productivity of firms [33]. According to Patti and Schifilliti, firms that complete digital transformation at a faster rate tend to contribute more to local economic growth. Conversely, firms with slower digital transformation make a limited contribution to economic growth, leading to a gap in regional economic development [34]. This digital divide arises from differences in digital transformation across industries and can be considered a horizontal digital divide effect. Regional economic development gaps resulting from the horizontal digital divide may increase the risk of relative income poverty among urban households.

Furthermore, the digital skills divide arises from differences in motivations for skills acquisition [35]. The use of computers and high-speed Internet by average families with

students has a negative impact on student performance [36]. This is mainly because families with low human capital rely more on smartphones for socializing and entertainment, while those with high human capital predominantly access information [37]. This results in a continuous high level of digital skills among households with high human capital, digital poverty among households with low human capital and a persistent widening of the digital skills gap among different households. These challenges pose a threat to the government's digital governance [38]. The government aims to address the digital access divide problem by building multiple digital platforms and upgrading intelligent devices [39]. However, the motivation of different households to apply digital skills deepens the adverse effects of the first-level digital access divide [40]. This generates a vertical second-level digital divide that exacerbates digital inequality among urban households and leads them to relative poverty.

**Hypothesis 2.** *The use of digital technologies in cities creates a horizontal and vertical divide, deepening the gap in economic development and digital inequality and triggering relative household poverty.*

*2.4. Digital Technology Development and Regional Income Distribution Inequality*

The economy is undergoing a digital transformation due to the increasing use of digital technologies in production and daily life. This transformation has a positive impact on economic growth and helps to reduce income inequality [41]. However, the existence of differences in digital technology proficiency among households of varying classes and characteristics creates a digital technology application divide that can impact income distribution in the digital economy. Research has shown that, in rural areas, intergroup digital inequality hinders household income growth and exacerbates inequitable income distribution [42]. Resource and spatial advantages lead digital platforms to prefer locating in cities [43]. However, there are spatial differences at the level of urbanization development between regions [44]. As a result, digital platforms tend to favor regions with higher levels of urbanization, leading to an inter-urban digital divide [45] and resulting in inequality in urban income distribution. Inequality in income distribution worsens inequality of opportunity in education and health among urban households, which increases the likelihood of relative poverty.

**Hypothesis 3.** *The adoption divide in digital technology, generated by digital technology development, can worsen urban income inequality and increase the likelihood of relative poverty in households.*

**3. Construction of the Model and Selection of the Index**

*3.1. Model Construction*

This paper constructs a benchmark model to examine the impact of digital technological progress on the relative poverty of urban households (as shown in Equation (1)).

$$poverty_{int} = \alpha_0 + \alpha_1 tec_{int} + \alpha_2 X_{it} + \varphi_n + v_t + \varepsilon_{it} \tag{1}$$

$poverty_{\text{int}}$ this section presents the family's relative poverty status in year t, $tec_{\text{int}}$ is the progress level of digital technology in the n area in year t, $X_i$ and a series of $i$ variables related to the family, $\phi_n$ and $v_t$ are, respectively, fixed effects of province and time, and $\varepsilon_{it}$ is the random error term. The coefficient $\alpha_1$ represents the influence of digital technology development on the relative poverty of urban households. If it is positive, it suggests that the advancement of digital technology may contribute to an increase in the relative poverty of urban households. Conversely, if it is a negative value, it implies a reduction in the relative poverty of urban households.

*3.2. Selection of Indicators*

**Interpreted variable:** family relative poverty. This paper focuses on urban households, and relative poverty mainly refers to the likelihood of a family falling into relative poverty.

Academics generally use relative poverty lines to determine whether families are in relative poverty. However, since this paper mainly studies the degree of deviation between urban household income and mainstream living conditions, it is reasonable to use a strong relative poverty line. Therefore, 60 percent of the international median income is used as the relative poverty line [46]. In the CFPS dataset and according to the per capita net income data in the questionnaire, taking into account the provincial economic development level gap and using a unified poverty line will result in measurement bias. Therefore, the sample must be taken according to the province and the provincial urban household per capita income sample must be taken at 60 percent and 50 percent. Moreover, if the family's per capita income numerical judgment is below the relative poverty line, it should be recorded as 1; otherwise, it should be recorded as 0.

Subsequently, the urban relative poverty line is calculated using the segmented function method (Equation (2)), which is based on Ravallion's Gini discounted mean method [47] (Equation (3)). To ensure the accurate measurement of poverty across provinces and urban areas, this paper calculates the annual Gini coefficients of provincial towns and cities. This results in a weak urban relative poverty line by province, which addresses the limitations of a uniform poverty line.

$$m_j^* = (1 - G_j)m \tag{2}$$

$$Z^{A+R} = \max(Z^A, (1 - \beta)Z^A + \beta m) \tag{3}$$

In Equation (2), m is the income mean, $m_j^*$ is the relative income, $G_j$ is the Gini coefficient; in Equation (3), $Z^A$ is the absolute poverty line of CNY 2300 per person per year implemented in China in 2011, $Z^{A+R}$ is the weak relative poverty line and the results of empirical analyses by Ravallion (2020) $\beta = 0.7$ are used to make the income elasticity [48]. The calculated weak relative poverty line is used to identify the poverty status of the sample households again for robustness testing.

**Core explanatory variable:** digital technology. The digital economy is developed with the support of a new generation of information technology. The essence of the digital economy is the progress of digital technology. The development of China's digital economy is government-led, and the digital technology development orientation of local governments is an important driving force for digital transformation. This paper utilizes a Python tool to identify the frequency of key digital technology words in provincial government work reports from 2010 to 2018. The ratio of the frequency of these words to the total word count is used to reflect the government's digital development orientation, which is recorded as tec. A higher proportion of key digital technology word frequency indicates a greater digital technology progress, while a lower proportion indicates a weaker one. The text presents a list of six key digital technology aspects: Big Data, blockchain, artificial intelligence, communication technology, Internet of Things and cloud computing. Table 1 shows the specific search phrases for each type of technical indicator. At the same time, the digital transformation orientation of listed companies in each province in recent years is represented by the number of digital invention patent applications, and this variable is recorded as tec2 for the robustness test.

**Mechanism variable:** Digital technology application divide. In this paper, we mainly examine how digital technology development affects the occurrence of relative poverty among families through different levels of digital technology application. The measurement of digital technology application includes the scope of digital technology application at the macro government level, and the scenario of digital technology application at the micro family level. At the macro level, the total word frequency ratio of digital technology applications in the government work reports of each province is used to indicate the specific application scenarios of digital technology in industry, service industry and government digital governance. Table 2 compares the differences in the application of key core technologies in urban poverty reduction among industries. The household economic questionnaire of the CFPS includes questions on whether a computer, mobile phone or email is used. A

"yes" answer is assigned a value of 1, while a "no" answer is assigned a value of 0. The sum of the three scores reflects the level of household digital technology adoption. The industrial and service sectors exhibit a higher degree of digital transformation, and the ratio of their digital technology application orientation highlights the difference in digital technology application among industries. This can be considered as the horizontal digital divide. The ratio of digital technology application between the government's digital governance and household levels highlights the difference in technology usage between management and target levels. This can explain the vertical digital divide between different levels.

**Table 1.** Classification and representation of key digital technologies.

| Key Technical Specifications | Phrase Listing |
| --- | --- |
| Big Data | Big Data, mixed reality, data warehousing, data mining, digital twins, virtual reality, heterogeneous data, augmented reality |
| Blockchain | differential privacy technology, distributed computing, blockchain, digital currency, smart contracts |
| Artificial intelligence | Robotics, machine learning, computer vision, decision aids, decision support systems, artificial intelligence, business intelligence, mining algorithms, virtual reality, intelligent robotics, intelligent technology, intelligent data analysis, intelligent algorithms, expert systems |
| Communications technology | 4G, 5G, 5G networks, 6G, communications, cybersecurity, satellite |
| Internet of Things | RFID, BeiDou Navigation System, positioning system, infrared sensor, laser scanner, RFID, Internet of Things, mobile Internet of Things, smart sensor |
| Cloud computing | EB-level storage, multi-party secure computing, brain-like computing, quantum computing, streaming computing, green computing, in-memory computing, cognitive computing, converged architectures, graph computing, IoT, information physical systems, billion level concurrency, cloud computing, cloud platforms |

**Table 2.** Digital technology applications by sub-sector.

| Industry Classification | Digital Technology Applications |
| --- | --- |
| Industry | Industrial digitization, industrial Internet, digital supply chain, driverless cars, smart wear, smart supply chain, smart home, smart transportation, smart production equipment, smart manufacturing |
| Service industries | B2B, B2C, C2B, NFC payment, O2O, third-party payment, e-commerce, Internet finance, Internet healthcare, fintech, open banking, quantitative finance, platform Internet, digital finance, digital economy, digital RMB, Netflix, cyber-entertainment, unmanned retailing, unmanned banking, information industry, mobile Internet, mobile payment, smart pension, smart healthcare, smart storage smart grid, smart environmental protection, smart home, smart customer service, smart energy, smart investment, smart culture and tourism, smart healthcare, autonomous driving |
| Government digital governance | Rural big data cloud platform, data center, digital service system, digital government, government service platform, government platform, government application system, smart city, smart village, intelligent computing center |

Income inequality. The main test is based on whether the development of the digital economy has led to the occurrence of relative poverty among urban households through the increase in inter-regional income inequality. The provincial Gini coefficients are usually used to reflect income inequality, and the provincial Gini coefficients for each type of income in the sample are calculated on the basis of per capita disposable household income, wage income and business income published by the CFPS.

**Control variables:** This paper's control variables include both household and individual levels, as shown in Table 3. Household-level control variables include household size, expressed as the number of people in the household, and dependency ratio, expressed as the number of people in the household who are older than 60 years old and younger than 16 years old, as well as the proportion of the total number of people in the household. The variables for household head characteristics include the square of their age, marital status, political affiliation and education level. Dummy variables are assigned for marriage and party membership. Education level is categorized as tertiary education and above (1), high school (2), junior high school (3), primary school (4) and illiterate or semi-literate and not attending school (5).

**Table 3.** Explanatory and descriptive statistics of the main variables.

| Classification | Variable Name | N | Mean | SD | Max | Min |
|---|---|---|---|---|---|---|
| | poverty1 | 15,260 | 0.269 | 0.444 | 1.000 | 0.000 |
| Dependent variable | poverty2 | 15,260 | 0.218 | 0.413 | 1.000 | 0.000 |
| | relapoverty | 15,260 | 0.256 | 0.436 | 1.000 | 0.000 |
| Independent variable | tec | 15,260 | 0.001 | 0.001 | 0.005 | 0.000 |
| | tec2 | 15,260 | 0.114 | 0.222 | 1.000 | 0.000 |
| | familysize | 15,260 | 3.296 | 1.180 | 10.000 | 0.000 |
| | fdr | 15,260 | 0.386 | 0.339 | 2.000 | 0.000 |
| Control variable | age2 | 15,260 | 2757 | 1539 | 8836 | 0.000 |
| | politic | 15,260 | 0.108 | 0.310 | 1.000 | 0.000 |
| | marriage | 15,260 | 0.803 | 0.398 | 1.000 | 0.000 |
| | edubackground | 15,260 | 1.717 | 1.751 | 6.000 | 0.000 |
| | app | 10,972 | 0.166 | 0.266 | 1.500 | 0.000 |
| | app1 | 14,188 | 0.000 | 0.000 | 0.004 | 0.000 |
| Mechanism variable | geni | 15,260 | 0.423 | 0.0510 | 0.574 | 0.248 |
| | wgeni | 11,897 | 0.407 | 0.0430 | 0.554 | 0.269 |
| | mgeni | 3327 | 0.564 | 0.122 | 0.851 | 0.000 |

*3.3. Data Processing and Descriptive Statistics*

The empirical analyses in this paper are primarily based on data from the 2010–2018 Chinese Family Tracking Survey Database and government work reports. The China Family Tracking Survey Database is a biennial family questionnaire survey conducted by Peking University. Since 2010, six rounds of surveys have been conducted, each consisting of four parts: the family economy questionnaire, the adult questionnaire, the child questionnaire and the family relationship questionnaire. This study utilized the survey components of the Household Economy Questionnaire and the Adult Questionnaire. As some indicators are missing in the 2020 dataset, only the 2010–2018 dataset is used. The data are cleaned to maintain the continuity of the sample households and indicators. The processing steps are as follows: firstly, households with missing or zero per capita disposable income are deleted to avoid invalid observations. Secondly, samples with less than five periods of household survey data are eliminated by matching household sample numbers in different observation years to ensure the continuity of observations. Thirdly, only urban households in the sample are retained according to the urban/rural classification code of the households. Finally, a final sample of 3052 urban households in 25 provinces is obtained. Table 3 shows the descriptive statistics of all variables.

## 4. Analysis of Experience

### 4.1. Analysis of Benchmark Results

Table 4 shows the results of the digitally oriented baseline estimates of relative poverty for urban households. The first two columns identify household relative poverty using a relative poverty line of 60% of median per capita disposable household income, while the last two columns use a relative poverty line of 50% of per capita disposable household income. A stepwise regression was used to analyze the data. No control variables were added to columns (1) and (3), while control variables were added to columns (2) and (4). The results indicate that digital technology development has a positive effect on the probability of relative poverty occurrence among urban households in all columns, and this effect is significant at the 5% level. Furthermore, the effect of digital technology development on household relative poverty decreases as the relative poverty line decreases with the addition of control variables. Specifically, if the relative poverty line is high, a 1% increase in digital technology development may increase the incidence of relative poverty among urban households by 15.64%. If the relative poverty line is low, a 1% increase in digital technology development may raise the probability of relative poverty among urban households by 14.03%. Hypothesis 1 is supported by the suggestion that digital technology development significantly contributes to relative poverty among urban households.

**Table 4.** The impact of digital technology development on the relative poverty of urban households.

| Variable | Poverty1 (1) | Poverty1 (2) | Poverty2 (3) | Poverty2 (4) |
|---|---|---|---|---|
| tec | 14.65 ** | 15.64 ** | 14.78 ** | 14.03 ** |
| | (2.70) | (2.88) | (2.86) | (2.70) |
| familysize | | 0.0197 *** | | 0.0163 *** |
| | | (5.64) | | (4.95) |
| fdr | | 0.0390 *** | | 0.0451 *** |
| | | (3.34) | | (4.11) |
| age2 | | 0.0000164 *** | | 0.0000168 *** |
| | | (6.02) | | (6.53) |
| politic | | −0.176 *** | | −0.143 *** |
| | | (−19.45) | | (−17.03) |
| marriage | | −0.0764 *** | | −0.0666 *** |
| | | (−7.79) | | (−7.19) |
| edubackground | | 0.0174 *** | | 0.0133 *** |
| | | (7.78) | | (6.26) |
| cons | 0.257 *** | 0.181 *** | 0.205 *** | 0.134 *** |
| | (43.95) | (11.21) | (37.29) | (8.87) |
| $R^2$ | 0.000506 | 0.0281 | 0.000595 | 0.0234 |
| N | 15,260 | 15,260 | 15,260 | 15,260 |

Note: The table includes estimated coefficients with t-values in brackets; ***, ** at the 1 per cent and 5 per cent, respectively.

Additionally, a range of control variables have been identified that increase the likelihood of relative poverty in urban households, including large family size, a heavy burden of dependents (both young and old) and an older head of household with lower levels of education. Conversely, the lack of information regarding the political profile and marital status of the head of household can reduce the likelihood of relative poverty, particularly if the head of household is a member of a political party. This membership can decrease the probability of relative poverty to some extent in urban households.

### 4.2. Robustness Test

4.2.1. Remeasurement of Variables

To avoid identification bias generated by the strong relative poverty line, this paper uses a segmentation function integrating absolute and relative poverty, drawing on Ravallion and Chen's method to measure the weak relative poverty line [18]. The average value

of urban household income is assigned a value based on the criteria of the weak poverty line, with values lower than the weak relative poverty line marked as 1 or 0. Secondly, the number of patent applications of listed companies in each province is used to represent tec2, which replaces the core explanatory variable for robustness test. The application volume is shown in Table 5. The (1)–(2) columns present the results of the analysis using the weak relative poverty line to identify the relative poverty of urban households. After adding control variables, the effect of digital technology development on the probability of relative poverty occurrence in households is positive and significant at the 1% level. The (3)–(4) columns replace the core explanatory variables with the patents for digital inventions in each province. The results also indicate that digital development orientation can increase the probability of relative poverty in urban households, which is consistent with the benchmark regression results.

**Table 5.** Results of robustness test—remeasurement of dependent and independent variables.

| Variable | Repoverty (1) | Repoverty (2) | Poverty1 (3) | Poverty1 (4) |
|---|---|---|---|---|
| tec | 19.21 *** | 21.65 *** | | |
| | (3.52) | (3.97) | | |
| tec2 | | | 0.0487 ** | 0.0360 * |
| | | | (2.91) | (2.15) |
| familysize | | 0.0242 *** | | 0.0181 *** |
| | | (7.01) | | (5.22) |
| fdr | | 0.0410 *** | | 0.0408 *** |
| | | (3.58) | | (3.50) |
| age2 | | 0.0000161 *** | | 0.0000164 *** |
| | | (6.00) | | (6.02) |
| politic | | −0.171 *** | | −0.175 *** |
| | | (−19.44) | | (−19.33) |
| marriage | | −0.0723 *** | | −0.0760 *** |
| | | (−7.48) | | (−7.75) |
| edubackground | | 0.0149 *** | | 0.0175 *** |
| | | (6.76) | | (7.81) |
| cons | 0.239 *** | 0.148 *** | 0.264 *** | 0.195 *** |
| | (41.12) | (9.25) | (65.46) | (12.82) |
| R-squared | 0.000901 | 0.0275 | 0.000597 | 0.0278 |
| N | 15,260 | 15,260 | 15,260 | 15,260 |

***, ** and * denote significant at the 1 per cent, 5 per cent and 10 per cent levels, respectively.

### 4.2.2. Model Settings

To avoid biased results when analyzing dummy variables with a linear choice model, a robustness test was conducted using the binary choice model Logit. The results are presented in Table 6. In the first two columns, the coefficient of the impact of digital technology development on the probability of the occurrence of relative poverty in urban households is positive and statistically significant when no control variables are added. After including the control variables in the last two columns, the impact coefficients remain positive and significant at the 5% level. This suggests that digital technology development increases the likelihood of relative poverty occurring among urban households. These findings are consistent with the results of the benchmark regression.

**Table 6.** Robustness tests—analyzed using Logit models.

| Variable | Poverty1 (1) | Poverty2 (2) | Poverty1 (3) | Poverty2 (4) |
|---|---|---|---|---|
| tec | 43.38 ** | 48.31 ** | 46.04 ** | 45.71 ** |
| | (2.75) | (2.94) | (2.84) | (2.69) |
| familysize | | | 0.0610 *** | 0.0571 *** |
| | | | (5.87) | (5.26) |
| fdr | | | 0.124 *** | 0.160 *** |
| | | | (3.45) | (4.24) |

**Table 6.** *Cont.*

| Variable | Poverty1 (1) | Poverty2 (2) | Poverty1 (3) | Poverty2 (4) |
|---|---|---|---|---|
| age2 | | | 0.0000501 *** | 0.0000568 *** |
| | | | (6.14) | (6.70) |
| politic | | | −0.651 *** | −0.597 *** |
| | | | (−15.41) | (−13.45) |
| marriage | | | −0.228 *** | −0.219 *** |
| | | | (−7.94) | (−7.36) |
| edubackground | | | 0.0520 *** | 0.0440 *** |
| | | | (7.95) | (6.44) |
| cons | −0.653 *** | −0.821 *** | −0.899 *** | −1.082 *** |
| | (−37.23) | (−44.72) | (−18.12) | (−20.86) |
| Pseudo $R^2$ | 0.0004 | 0.0005 | 0.0259 | 0.0238 |
| N | 15,260 | 15,260 | 15,260 | 15,260 |

*** and ** denote significant at the 1 per cent, and 5 per cent levels, respectively.

### 4.2.3. Clustering to Different Levels

To further exclude the interference of household, year and province factors in the estimation results, robustness tests were conducted using clustered robust standard errors. The effects of the core explanatory variables were tested by clustering to the household and year level, the province and year level and the household, province and year level, respectively. The results are shown in the Table 7, and the coefficients on the impact of digital technology development on the relative poverty of urban households are almost unchanged, with only a slight change in significance, and statistically significant overall, after considering the clustered robust standard errors at different levels in columns (1)–(3), respectively. This suggests that digital technology development exacerbates the probability of urban household relative poverty, consistent with the results of the benchmark regression.

**Table 7.** Robustness tests—clustering to different levels.

| Variable | Poverty1 (1) | Poverty1 (2) | Poverty1 (3) |
|---|---|---|---|
| tec | 15.64 *** | 15.64 * | 15.64 *** |
| | (2.86) | (1.66) | (2.86) |
| familysize | 0.0197 *** | 0.0197 *** | 0.0197 *** |
| | (5.56) | (4.37) | (5.56) |
| fdr | 0.0390 *** | 0.0390 ** | 0.0390 *** |
| | (3.31) | (2.80) | (3.31) |
| age2 | 0.0000164 *** | 0.0000164 *** | 0.0000164 *** |
| | (5.97) | (4.39) | (5.97) |
| politic | −0.176 *** | −0.176 *** | −0.176 *** |
| | (−19.36) | (−16.54) | (−19.36) |
| marriage | −0.0764 *** | −0.0764 *** | −0.0764 *** |
| | (−7.71) | (−7.90) | (−7.72) |
| edubackground | 0.0174 *** | 0.0174 *** | 0.0174 *** |
| | (7.70) | (5.11) | (7.70) |
| cons | 0.181 *** | 0.181 *** | 0.181 *** |
| | (11.08) | (8.25) | (11.09) |
| family-year | YES | NO | NO |
| province-year | NO | YES | NO |
| province-family-year | NO | NO | YES |
| R-squared | 0.0281 | 0.0281 | 0.0281 |
| N | 15,260 | 15,260 | 15,260 |

***, ** and * denote significant at the 1 per cent, 5 per cent and 10 per cent levels, respectively.

### 4.2.4. Further Exclusion of Policy and Other Factors

As some low-income households receive government subsidies such as living allowances, assistance payments, unemployment benefits and price subsidies, the sample is assigned a value of 1 based on the results of the CFPS household economic questionnaire on "receiving government subsidies". The CFPS household economic questionnaire results were used to determine whether participants received government subsidies. A value of 1 was assigned to those who received subsidies and 0 to those who did not. This was done to eliminate any interference from poverty assistance policies in the results. During the sample period, the trade relationship between China and the United States became tense. This may inhibit regional economic growth and lead to relative poverty in urban households. Therefore, the change in the trade relationship between China and the United States was added to the model as a control variable to exclude the interference of external factors.

Table 8 displays the results of the regression with the gradual addition of two control variables. The relationship between whether or not one receives government subsidies and the relative poverty of urban households is positive and significant at the 1% level in columns (1) and (3). Furthermore, the effect of digital technology development on the occurrence of relative poverty among urban households remains positive and statistically significant. Columns (2) and (4) present the estimation results of including the U.S.–China trade war in addition to the previous columns. The results suggest that U.S.–China trade tensions have a positive impact on the relative poverty of urban households. Furthermore, the effect of digital technology development as a dependent variable on the relative poverty of urban households is consistent with the benchmark regression results. This suggests that the impact of digital technology development on relative poverty in urban households remains strong even when accounting for external factors such as policy and political relationships.

**Table 8.** Robustness tests—excluding policy and other factors from interference.

| Variable | Poverty1 (1) | Poverty1 (2) | Poverty2 (3) | Poverty2 (4) |
|---|---|---|---|---|
| tec | 14.50 ** | 14.67 ** | 13.10 * | 13.27 ** |
| | (2.71) | (2.74) | (2.56) | (2.60) |
| familysize | 0.0202 *** | 0.0185 *** | 0.0169 *** | 0.0152 *** |
| | (5.92) | (5.16) | (5.24) | (4.48) |
| fdr | 0.0428 *** | 0.0399 *** | 0.0488 *** | 0.0458 *** |
| | (3.75) | (3.45) | (4.54) | (4.21) |
| age2 | 0.0000146 *** | 0.0000144 *** | 0.0000153 *** | 0.0000151 *** |
| | (5.48) | (5.40) | (6.05) | (5.97) |
| politic | −0.151 *** | −0.150 *** | −0.119 *** | −0.119 *** |
| | (−16.89) | (−16.79) | (−14.42) | (−14.32) |
| marriage | −0.0688 *** | −0.0685 *** | −0.0599 *** | −0.0596 *** |
| | (−7.19) | (−7.16) | (−6.60) | (−6.57) |
| edubackground | 0.0143 *** | 0.0149 *** | 0.0103 *** | 0.0109 *** |
| | (6.47) | (6.67) | (4.93) | (5.18) |
| subside | 0.208 *** | 0.209 *** | 0.193 *** | 0.195 *** |
| | (23.71) | (23.76) | (22.96) | (23.00) |
| tradewar2 | | 0.0501 * | | 0.0500 * |
| | | (1.65) | | (1.77) |
| cons | 0.126 *** | 0.129 *** | 0.0832 *** | 0.0858 *** |
| | (7.99) | (8.09) | (5.62) | (5.75) |
| R-squared | 0.0697 | 0.0699 | 0.0651 | 0.0653 |
| N | 15,259 | 15,259 | 15,259 | 15,259 |

***, ** and * denote significant at the 1 per cent, 5 per cent and 10 per cent levels, respectively.

### 4.2.5. Treatment of Endogenous Problems

This paper explores the potential causal relationship between digital technology development and relative poverty. The government's top–down push for digital technological

advancement may threaten urban relative poverty, while an increase in the probability of urban relative poverty can also put heavy pressure on local governments to pay more attention to digital economic development. To mitigate the impact of this endogenous issue on the outcomes, this paper employs an instrumental variables method. To satisfy the relevance and exogeneity requirements of instrumental variables, the educational data of the longest-serving provincial party secretaries in each province since 2000 is selected as an instrumental variable. This will have an impact on the local digital technology orientation and no effect on the current relative poverty of urban households. The language used is clear, objective and value-neutral, with a formal register and precise word choice. The sentence structure is simple and logical, with causal connections between statements. The text is free from grammatical errors, spelling mistakes and punctuation errors. No changes in content were made. Estimation was conducted using two-stage least squares. The results are presented in Table 9. Columns (1) and (2) test the effect of digital technology development on the relative poverty of urban households at the high relative poverty line using the academic qualifications of the longest-serving provincial party secretary as the instrumental variable. Columns (3) and (4) test the effect of digital technology development on the relative poverty of urban households at the low relative poverty line. The Wald F-statistic and Sargen statistic results indicate that the instrumental variables are valid and not weakly instrumented. The effect of digital technology development on the probability of urban household relative poverty remains positive and consistent with the benchmark regression.

**Table 9.** Treatment of endogenous problems.

| Variable | Tec The First Stage (1) | Poverty1 The Second Stage (2) | Tec The First Stage (3) | Poverty2 The Second Stage (4) |
|---|---|---|---|---|
| Tec | | 59.443 *** | | 59.24 *** |
| | | (1.77) | | (3.22) |
| eduofgovernor2 | 0.0002 *** | | 0.0002 *** | |
| | (34.65) | | (34.65) | |
| Constant | 0.0006 *** | 0.134 *** | 0.0006 *** | 0.0856 *** |
| | (23.06) | (5.14) | (23.06) | (3.52) |
| Observations | 15,260 | 15,260 | 15,260 | 15,260 |
| R-squared | | 0.2866 | | 0.2321 |
| Wald F statistic | 1200.60 | | 1200.60 | |
| Sargen statistic | 0.000 | | 0.000 | |

*** denote significant at the 1 per cent.

### 4.3. Heterogeneity Analysis

4.3.1. Differences in the Vulnerability of Households to Poverty in Areas with Different Economic Characteristics

The emergence of relative poverty within families is closely related to regional economic development and is also strongly correlated with family characteristics. The sample is divided into economically developed and less economically developed regions based on the total economic volume of the provinces in which it is located. The impact of digital technology development on the relative poverty of households in regions with different levels of economic development is then analyzed, as shown in Table 10. The data in columns (1)–(2) indicate that digital technology development increases the susceptibility of urban households in economically developed regions to relative poverty, while it has a significant poverty-reducing effect on urban households in less economically developed regions.

Secondly, based on the stage characteristics of China's urbanization development, areas with urbanization rates above 60% are classified as high urbanization level areas, while those below 60% are classified as low urbanization level areas. This classification was used to investigate the impact of digital technology development on relative poverty in areas with different urbanization levels. The results are presented in columns (3) and (4) of the table. In areas with high urbanization, digital technology development worsens the incidence of relative poverty among urban households. Conversely, in areas with low

urbanization, digital technology development reduces the incidence of relative poverty among urban households.

**Table 10.** Differences in the impact on households in areas with different economic characteristics.

| Variable | Higheonomic (1) | Loweonomic (2) | Highurban (3) | Lowurban (4) |
|---|---|---|---|---|
| tec | 30.82 *** | −15.49 ** | 29.52 *** | −13.88 ** |
| | (9.60) | (−3.24) | (7.12) | (−2.97) |
| familysize | 0.0110 *** | 0.00665 * | −0.00273 | 0.0224 *** |
| | (4.92) | (2.44) | (−1.09) | (7.64) |
| fdr | 0.0348 *** | −0.0159 | 0.0372 *** | 0.00187 |
| | (4.64) | (−1.81) | (4.13) | (0.20) |
| age2 | 0.0000113 *** | −0.0000399 | 0.0000131 *** | 0.0000323 |
| | (6.48) | (−0.20) | (6.45) | (1.48) |
| politic | −0.0551 *** | −0.0858 *** | −0.0773 *** | −0.0991 *** |
| | (−10.12) | (−13.65) | (−11.47) | (−14.59) |
| marriage | −0.0257 *** | −0.0469 *** | −0.0388 *** | −0.0376 *** |
| | (−4.28) | (−5.96) | (−5.25) | (−4.67) |
| edubackground | −0.000142 | 0.0134 *** | 0.00255 | 0.0149 *** |
| | (−0.10) | (7.57) | (1.52) | (7.94) |
| cons | −0.00361 | 0.156 *** | 0.0863 *** | 0.0949 *** |
| | (−0.37) | (12.01) | (7.22) | (7.27) |
| R-squared | 0.0173 | 0.0156 | 0.0194 | 0.0203 |
| N | 15,260 | 15,260 | 15,260 | 15,260 |

***, ** and * denote significant at the 1 per cent, 5 per cent and 10 per cent levels, respectively.

4.3.2. Differences in Vulnerability to Poverty across Household Characteristics

First of all, according to the education level of the head of the household, households with a head who has tertiary education or above are classified as high-knowledge-level households and assigned a value of 1. Similarly, households with a head who has senior high school, junior high school, primary school, illiterate or semi-illiterate education are classified as low-knowledge-level households. The household is classified as having a low level of knowledge and is assigned a value of 0. The cross-multiplication term of the head of the household's education level with the digital technology indicates that urban households of different knowledge levels adapt to digital development. The effect on the occurrence of relative poverty is tested and the results are shown in Table 11. When compared to households with a high level of knowledge, low knowledge households are more likely to experience relative poverty when using digital technology. This is possibly due to their lower sensitivity to policy orientation and difficulty in adapting to the needs of digital economy development. As a result, they are at a higher risk of falling into digital poverty.

Secondly, a value of 1 is assigned to families with a dependency ratio higher than 0.5, and 0 is assigned to all others. Families are distinguished based on their dependency ratios, which can be high or low. This study measures the understanding of the government's intention of digital technology development among families with different dependency ratios by using cross-multipliers of dummy variables and analyzing the impact on the relative poverty of urban families. The results, as shown in Table 11's (3) and (4) columns, indicate that families with high dependency ratios face more difficulties in adapting to the government's digital development plans. Families with higher dependency ratios are less likely to comply with the government's digital development goals and tend to allocate more human and financial resources to improve their digital skills. As a result, they are more likely to be classified as relatively poor families and become a new type of digital poverty group in towns and cities compared to those with lower dependency ratios. Overall, enhancing the government's digital technology development can increase the likelihood of relative poverty for households with heads of low education levels, while decreasing it for households with high dependency ratios.

**Table 11.** Differences in poverty vulnerability by household characteristics.

| Variable | Poverty1 (1) | Poverty1 (2) | Poverty1 (3) | Poverty1 (4) |
|---|---|---|---|---|
| hedu | −8.010 (−1.29) | | | |
| ledu | | 45.36 *** (5.37) | | |
| hfdr | | | 27.83 *** (3.56) | |
| lfdr | | | | 1.858 (0.30) |
| familysize | 0.0177 *** (5.08) | 0.0192 *** (5.55) | 0.0192 *** (5.54) | 0.0184 *** (5.29) |
| fdr | 0.0426 *** (3.66) | 0.0371 ** (3.18) | 0.00980 (0.67) | 0.0438 *** (3.36) |
| age2 | 0.0000164 *** (6.03) | 0.0000166 *** (6.13) | 0.0000161 *** (5.91) | 0.0000164 *** (6.02) |
| politic | −0.174 *** (−19.06) | −0.172 *** (−18.89) | −0.176 *** (−19.40) | −0.175 *** (−19.33) |
| marriage | −0.0767 *** (−7.83) | −0.0765 *** (−7.81) | −0.0753 *** (−7.67) | −0.0767 *** (−7.82) |
| edubackground | 0.0159 *** (6.08) | 0.00763 ** (2.64) | 0.0175 *** (7.83) | 0.0175 *** (7.83) |
| cons | 0.207 *** (12.50) | 0.198 *** (13.11) | 0.197 *** (13.05) | 0.196 *** (11.97) |
| R-squared | 0.0276 | 0.0297 | 0.0284 | 0.0275 |
| N | 15,260 | 15,260 | 15,260 | 15,260 |

***, ** denote significant at the 1 per cent, 5 per cent levels, respectively.

## 5. Analysis of the Influence Mechanism

A digital development orientation led by the government will inevitably result in the application of advanced production technologies, leading to the digital transformation of market players, including businesses and households. The attributes of enterprises in different industries vary greatly, which may result in varying degrees of digital transformation and an inter-industry gap in the application of digital technology. The level of digital technology application among household members is determined by their specificity. Low digital technology application levels may result in digital poverty, making such households the primary target of government governance. The government is responsible for building digital platforms and implementing digital governance. However, households may struggle to adapt to the government's digital governance in a timely manner, resulting in a vertical digital technology divide between the management level and the target level. Furthermore, the digital development orientation can affect inter-regional income inequality. This paper explores the impact of digital technology on the relative poverty of urban households, focusing on the digital technology adoption divide and regional income inequality.

### 5.1. Digital Technology Application Divide

Digital technology orientation creates a favorable environment for the application of digital technology in both production and daily life. The industrial and service sectors have made significant progress in digital transformation, with the former showing a higher ratio of digital technology application compared to the latter. This highlights the existing gap in digital technology application between industries. On the one hand, the construction of digital platforms and the increased application of digital technology at the government level have led to an improvement in digital governance. On the other hand, whether families can adapt to the digital transformation trend of the government and improve their digital skills in daily life depends more on the inherent characteristics of the family members. Therefore, the ratio of the government's digital governance level to the family's digital skill

application level can be used to express the management level and the target level of the vertical digital divide. This section examines how digital development orientation affects the divide in vertical and horizontal digital skills adoption, and how this divide leads to relative household poverty (Table 12).

**Table 12.** Mechanisms for influencing horizontal and vertical divides in the use of digital technologies.

| Variable | App (1) | Poverty1 (2) | App1 (3) | Poverty1 (4) |
|---|---|---|---|---|
| tec | 80.28 *** | | 0.0732 *** | |
| | (22.05) | | (15.14) | |
| app | | 0.0348 * | | |
| | | (2.16) | | |
| app1 | | | | 107.1 *** |
| | | | | (7.66) |
| familysize | −0.0219 *** | 0.0194 *** | −0.000218 *** | 0.0197 *** |
| | (−10.28) | (4.62) | (−0.87) | (5.48) |
| fdr | 0.00150 | 0.0411 ** | 0.0000641 *** | 0.0196 |
| | (0.17) | (3.00) | (8.17) | (1.64) |
| age2 | 0.000322 *** | 0.0000214 *** | 0.000551 ** | 0.0000121 *** |
| | (1.57) | (6.70) | (2.91) | (4.23) |
| politic | 0.00207 | −0.178 *** | −0.0000582 *** | −0.159 *** |
| | (0.26) | (−16.54) | (−9.11) | (−17.48) |
| marriage | −0.0163 * | −0.0735 *** | −0.000115 | −0.0618 *** |
| | (−2.38) | (−6.39) | (−1.60) | (−6.07) |
| edubackground | −0.00665 *** | 0.0198 *** | 0.0000130 *** | 0.0182 *** |
| | (−4.61) | (7.58) | (8.41) | (7.84) |
| cons | 0.176 *** | 0.177 *** | 0.0000823 *** | 0.168 *** |
| | (15.35) | (9.54) | (7.44) | (10.69) |
| R-squared | 0.0742 | 0.0318 | 0.0526 | 0.0307 |
| N | 10,972 | 10,972 | 14,188 | 14,188 |

***, ** and * denote significant at the 1 per cent, 5 per cent and 10 per cent levels, respectively.

Table 12 displays the results. Column (1) indicates that digital technology development widens the horizontal digital technology adoption divide between industries, mainly because industry is much more digitally transformed than services. Column (2) demonstrates that a 1% increase in the horizontal adoption divide of digital technology leads to a 0.0348% increase in the probability of relative poverty occurring in towns and cities. This result is significant at the 10% level. The adoption of urban digital technology for poverty reduction fails, and the inter-industry digital technology divide creates gaps in industry development. This can impact the income disparity of the urban labor force working in different industries and exacerbate the likelihood of urban households falling into relative poverty. In contrast, column (3) demonstrates a positive correlation between digital technology development and the vertical inter-sector and inter-household divide in digital technology adoption. Specifically, for every percentage point increase in digital technology progress, the vertical digital technology adoption divide may widen by 0.0732 per cent. Column (4) demonstrates that a rise in the vertical digital divide results in a significant increase in the likelihood of urban households experiencing relative poverty. Specifically, a 1% increase in the vertical digital divide leads to a 107.1% increase in the probability of urban households being relatively poor. Specifically, a 1% increase in the vertical digital divide leads to a 107.1% increase in the probability of urban households being relatively poor. It is evident that digital technology development may increase the likelihood of urban households experiencing relative poverty due to both horizontal and vertical divides in digital technology application. Of particular concern is the impact of the vertical divide in digital technology application on urban households' relative poverty. If households fail to adjust to the digital lifestyle promptly, they may fall into the relative poverty trap.

*5.2. Inequality in Income Distribution*

Digital technology is a significant factor in driving total factor productivity. The digital divide resulting from differences in digital technology development not only widens the gap in digital application between industries but also creates a divide in digital application at various levels, resulting in productivity differences and a loss of efficiency in government management. The income inequality of industries widens due to differences in total factor productivity, while the inability of some families in need to come out of poverty reflects the loss of government management effectiveness. To analyze the negative impact of the digital technology application gap, we use the Gini coefficient to test the impact of digital technology development on the relative poverty of urban households from the perspective of income inequality. This includes the inter-household Gini coefficients of household disposable income, wage income and business income. Only the impact of digital technology development on income inequality is examined, as shown in Table 13. Income inequality is a direct cause of households falling into relative poverty.

**Table 13.** Mechanisms affecting income inequality.

| Variable | Gini (1) | Wgini (2) | Mgini (3) |
|---|---|---|---|
| tec | 5.274 *** | 9.330 *** | 8.867 ** |
| | (5.43) | (14.44) | (2.84) |
| familysize | 0.00617 *** | 0.00211 *** | −0.0122 *** |
| | (16.77) | (5.59) | (−6.24) |
| fdr | −0.00632 *** | 0.00210 | −0.00354 |
| | (−4.52) | (1.50) | (−0.48) |
| age2 | 0.0000055 | 0.000000976 ** | 0.0000319 |
| | (1.77) | (3.12) | (−0.18) |
| politic | −0.00984 *** | −0.00434 ** | 0.0137 |
| | (−7.41) | (−3.21) | (1.58) |
| marriage | 0.00357 ** | 0.00373 *** | 0.0198 ** |
| | (3.28) | (3.54) | (3.07) |
| edubackground | −0.00179 *** | 0.00338 | −0.00583 *** |
| | (−7.49) | (−0.14) | (−5.22) |
| cons | 0.400 *** | 0.386 *** | 0.598 *** |
| | (204.21) | (208.55) | (57.47) |
| R-squared | 0.0323 | 0.0254 | 0.0313 |
| N | 15,260 | 11,897 | 3327 |

***, ** denote significant at the 1 per cent, 5 per cent levels, respectively.

The table above displays the impact of digital technology development on the Gini coefficient of total income, wage income and business income in columns (1), (2) and (3), respectively. The results indicate a positive and statistically significant correlation between digital technology development and all three coefficients. A one-percentage-point increase in digital technology development as the greatest impact on wage income inequality, leading to a 9.33 percentage point increase. Business income inequality is about 8.87 percentage points greater as a result of digital technology progress. The impact on overall income inequality is slightly smaller, at 5.27 percentage points. The digital technology development has led to an increase in income distribution inequality, which poses a potential risk for the occurrence of relative poverty among urban households.

## 6. Conclusions and Suggestions

*6.1. Conclusions and Contribution*

The goal of achieving common prosperity has been proposed, and the development of the digital economy has been identified as an important means of promoting a sustainable economic dynamic. While previous studies have primarily examined the impact of digital economic development on reducing rural poverty, this paper moves the perspective to the urban side and focuses on the impact of digital technology development on the relative

poverty of urban households, which not only expands the scope of the research on relative poverty, but also reveals the external mechanism of the impact of digital technology development on the relative poverty of urban households in terms of the digital technology application gap dimension, enriching the research content of relative poverty in the new era. Furthermore, this study examines the relationship between poverty and income inequality and explores the issue of income inequality resulting from the gap in digital technology application. This forms a progressive logical system that delves into the distributional effects of digital technology at the microfamily level.

This study's results indicate that the intensification of digital technology development increases the likelihood of relative poverty in urban areas. Specifically, digital technology-induced poverty is prevalent among households in economically developed and highly urbanized areas with low levels of education among the heads of household and high household dependency ratios. The analysis of impact mechanisms reveals that the development of digital technology affects the relative poverty of urban households due to the horizontal and vertical divide in digital technology adoption, as well as to the unequal distribution of income among households.

### 6.2. Policy Implications

All of the research results offer insights for promoting a sustainable economic dynamic in China. First of all, urban households have more complex characteristics and require greater attention regarding relative poverty. In economically developed provinces, low-income households should be cautious of digital poverty resulting from economic poverty and the risk of falling into a cycle of relative poverty. Secondly, the proliferation of key digital technology applications should be balanced to prevent the emergence of horizontal and vertical digital divides. In regions with low initial levels of digital technology application, industries should undergo digital transformation in a layered manner. This will accelerate the promotion of digital technology applications and help to alleviate the digital development gap between regions. Simultaneously, the central government should enhance the vertical digital assistance mechanism to aid localities and families in a timely manner. This can be achieved by improving the vertical penetration performance of digital technology through the provision of training courses and organizing technology exchange conferences. Additionally, stimulating and maintaining a digital-friendly atmosphere in families through economic and cultural means can promote the establishment of an intrinsic feeder mechanism in families. This will help to narrow the negative impact of the vertical digital divide on the income increase in urban families. Thirdly, it is important to consider the external environment when developing a digital economy. It is necessary to guide the development of small and medium-sized digital enterprises in a reasonable and orderly manner, while also optimizing the entrepreneurship and employment market environment. The government's focus on digital technology is likely to lead to increased competition within the industry. Therefore, it is important to guard against the potential monopoly effect of platforms, which could trigger competitive chaos and damage the ecological environment of digital economic development. We will provide support and assistance to new small and medium-sized digital enterprises to enable them to fully utilize their job creation function. Additionally, we will assist individual entrepreneurs in upgrading their digital skills in a timely manner and carrying out their business dealings with digital platforms smoothly. This will lay a solid foundation for stable digital employment in various regions.

### 6.3. Limitations and Suggestions for Future Research

Measuring the development of digital technology is a challenging task. This study employs the prevalent text analysis method and the number of patent applications related to digital technology to characterize its development. Additionally, it explores the impact of digital technology on the income of urban residents. Future research should aim to reflect the development of digital technology with specific and detailed indicators. The focus should be on the penetration of digital technology into the field of sustainable economic

development and its far-reaching impact on various production factors and economic activity participants. Simultaneously, numerous digital technology factors impact the relative poverty of urban residents. Future research should focus on exploring micro-level digital technology factors.

**Author Contributions:** Methodology, S.J.; Formal Analysis, S.J.; Investigation, S.J.; Resources, S.J.; Writing—Original Draft, S.J.; Writing—Review and Editing, S.J.; Project Administration, S.J.; Funding Acquisition, F.D. All authors have read and agreed to the published version of the manuscript.

**Funding:** This research was funded by the Research on the Path of "Industry Aid to Xinjiang" in the Problem of Regional Coordinated Development Mechanism of the China National Social Science Foundation Program (18BJL083). Autonomous region Universities Basic research Businesses Funding for research projects (XJEDU2024J009).

**Institutional Review Board Statement:** This study was conducted according to the guidelines of the Declaration and approved by the Ethics Committee of the Deanship of Scientific Research.

**Informed Consent Statement:** All subjects gave their informed consent for inclusion before they participated in this study. This study was conducted in accordance with the declaration of the authors' universities, and the protocol was approved by the Ethics Committee.

**Data Availability Statement:** Data will be available upon request.

**Conflicts of Interest:** The authors declare no conflicts of interest.

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
