# Peer review of "Research on Sustainable Economic Dynamics: Digital Technology Development and Relative Poverty of Urban Households"

_sustainability, doi:10.3390/su16083407_

Round 1

Reviewer 1 Report

Comments and Suggestions for Authors

The abstract did not describe enough clearly the very exciting content of the paper.

It is positive that it considers the relative poverty of urban households as opposed to the relative poverty of rural families. Model constructing was very successful in this paper.

First hypothesis: Digital adoption divide and relative urban povertyHypothesis 1 is supported by the suggestion that the development of digital technology significantly contributes to relative poverty among urban households.

Second hypothesis: Development of digital technology and inequality in regional distribution of incomeThe development of digital technology increases the vulnerability of urban households in economically developed regions to relative poverty, while it has a significant poverty-reducing effect on urban households in less economically developed regions.

Third hypothesis: The divide in digital technology adoption, created by the development of digital technology, can worsen income inequality in urban areas and increase the probability of relative poverty in households.This paper explores the impact of digital technology on the relative poverty of urban households, focusing on digital technology adoption divides and re-regional income inequality.

If households fail to adapt to a digitized lifestyle soon, they could fall into the trap of relative poverty. Urban households have more complex characteristics and require greater attention to relative poverty. In economically developed provinces, households with low incomes should be careful because of digital poverty, which is a consequence of economic poverty and the risk of falling into the cycle of relative poverty. In addition, the proliferation of key digital technology applications needs to be balanced to prevent the emergence of horizontal and vertical digital divides. In regions with low initial levels of digital technology adoption, industries should undergo digital transformation in a layered manner.

It will speed up the promotion of digital technology adoption and help reduce the digital development gap between regions. At the same time, the central government should improve the vertical mechanism of digital assistance to help local communities and families in a timely manner. Encouraging and maintaining a digitally adapted atmosphere in families through economic and cultural means can promote the establishment of an intrinsic feeding mechanism in families. This paper will help narrow the negative impact of the vertical digital divide on increasing the income of urban families.

Reviewer 2 Report

Comments and Suggestions for Authors

Based on an econometric analysis of real data, this paper discusses the tendency for the advancement of digital technology to increase disparities among residents and industries in urban China. The analytical methodology is robust, and the interpretation of the results and recommendations for the future of society are convincing.

However, the following points require improvement and explanation

The explanations of what exactly tec1 and tec2 are, which are listed as important explanatory variables, are unclear. Is the tec1 "the ratio of the frequency of these words to the total word count (Line285)"?  If it is wright, it is unclear why the sample size is 15260. Is the tec2 the responses to the household economic questionnaire of the CFPS (Line 307)?  If there is more than one question, the maximum value should be the number of questions, not equal to 1.

The survey seems to use data from 2010-2018. covid-19 should have had a significant impact on the digital society, which needs to be discussed in the paper.

Minor points.

Line243 is (1.1) not (0.1)

Reviewer 3 Report

Comments and Suggestions for Authors

I congratulate the authors on an excellent choice of research topic, on originality, with a focus on rigorous methodology, but not on the explained theoretical implications and contributions to policies, as well as on the theoretical foundations in the Introductory part. I think you need to make corrections, which I will elaborate on below.

In the Abstract of the paper, you did not state which data processing methods you used and very briefly the theoretical and policy implications of your research. In the introductory part, you stated several important facts that cannot be verified, because you did not indicate the sources of the literature. For example, on page 1, rows 33-35 you wrote the following: „Meanwhile, the economic development model that utilises digital technology as its core driver has made significant progress during the epidemic due to its interconnected  production sharing and convenient consumption“.

Further, you specified in rows 36-39, again without adequate citation following:

The digitalisation of industries and the development of digital industrialisation have created obstacles to the employment of low-skilled groups in cities. This has led to an increase in household unemployment, difficulties in securing income, and a gradual highlighting of the problem of relative poverty“.

These two statements, without confirmation from the literature, are quite controversial and contradictory. Does this mean that we should stop the digital transformation because it will increase the relative power of low skilled human resources in urban households.

On page 2, rows 45-47 you stated following:

„To address employment problems and reduce relative poverty while promoting sustainable development of local economies, the Chinese government has issued the 14th Five-Year Plan for Employment Promotion“.

Another sentence without proper citation! And there are more below, which you should correct yourself until the end of the text.

On the same page, rows 50-56:

„Adjustment of employment-stabilising policies and measures to promote development and safeguard people's livelihoods", etc., advocating that workers adapt in a timely manner to the needs of the development of a digital China, an intelligent China, a healthy China, and a premanufacturing country, and that they participate in skills training to upgrade their skills, and that they achieve the sustainability of their individual income-generating efforts by means of specialised business operations  and flexible, multi-channel employment“.

This sentence is too long, vague and confusingly written! On the same page, rows 57-66:

 Simultaneously, China's digital technologies, including big data, cloud computing, blockchain, artificial intelligence, and 5G, have become the core driving force for sustainable economic development. These technologies have penetrated various industries, bringing about changes in production. Digital technology has transformed traditional factors of production and enhanced the social changes resulting from the inclusion of data as a new type of factor of production. This is reflected in the digital and intelligent features of production equipment, which demand higher levels of digital skills and knowledge from workers. Additionally, workers have benefited from the development of digital technology, increasing their use of digital products and improving their application of digital skills. This lays the foundation for a potential shift in existing income distribution“.  All above written sentences are taken from or have paraphrased certain quotes, which you did not mention in this part of the paper.

Furthermore, you stated the following: „The technology is categorized into two types: key core technology and general technology“. What is key core technology and what is general technology? You need to define key terms from adequate sources.

 In the introduction, contributions of the study should be explicitly identified and briefly elaborated. For instance, the first contribution is ____The second contribution is _____. The third contribution is _____. These contributions should be explained in few sentences for each contribution.

 It is customary to describe the structure of the continuation of the paper at the end of the Introduction. E.g. The rest of this study is organized as follows. In the next section, we provide a review of the literature related to digital transformation and innovation in SMEs. The following section presents the methodology by explaining the sample, model specification and the empirical strategy. The next section presents empirical results followed by the main conclusions, limitations and ideas for future research.

Technically, you should adapt the citation to the journal's requirements (https://www.mdpi.com/journal/sustainability/instructions).

  You have written the rest of the paper very correctly except for the last part, which lacks the Discussion part in which you should review the findings and puts them into the context of the overall research as well as compare the results of your research with similar research. Finally, you Conclusion session is a little bit messy.  I suggest you introduce subheadings in the Conclusion with which you will systematize the concluding considerations. These can be the following subheadings:

6. Conclusions and Implications

6.1. Theoretical contributions

6.2. Policy and managerial implicatons

6.3.  Limitations and suggestions for future research

Your research makes at least two of three above mentioned contributions, you just need to elaborate them in detail and carefully. With major corrections made, your research will capture the attention of the academic and professional public.

Comments on the Quality of English Language

No comments.

Round 2

Reviewer 2 Report

Comments and Suggestions for Authors

Regarding my Quesion1, the authors' explanation made sense to me. However, unless this is explained in the text, the reader will not understand it as well as I do. Where exactly in the text does this explanation appear? There does not appear to be any mention of tec1 or tec2 in the text.

Equation (1.1) instead of equation (1.0) in Line 244

The same sentence is repeated three times in line 287-291.

Author Response

Thank you for reviewing my revised draft again. Your feedback has helped me improve the quality of my article. In response to your suggestion, I have revised the problems in the article. We hope these modifications meet your requirements.

Answer to Question 1: Thank you for your question, it is true that the previous manuscript was not clearly marked. In this paper, the core explanatory variable is named tec, and relevant descriptions are in lines 296-300. tec2 is used for robustness testing, as described in robustness Testing 4.2.1 in the previous manuscript. To give the reader a clearer picture of the description of the core explanatory variables, we have added tec2 at the end of the explanatory paragraph of the core explanatory variables. For details, see lines 301-303 in revised Draft 2. At the same time, the statement in robustness test 4.2.1 isThank you for your offer. In the sixth part of the paper, I add the limitations of this study and suggestions for future research. See lines 686-696 for details, with all changes in blue font.Thank you for your offer. In the sixth part of the paper, I add the limitations of this study and suggestions for future research. See lines 686-696 for details, with all changes in blue font. more concise than before, specifically see lines 400-402. The author's text for all changes is shown in blue font.

Answer to Question 2: Thank you for reminding me, I have corrected the error of formula number in line 251.

Answer to Question 3: Thank you very much for your careful review. I have removed the duplicates. See lines 294-296 for details.

Reviewer 3 Report

Comments and Suggestions for Authors

Congratulations authors for the significant improvement in the quality of the paper. In Conclusion, I suggest you to add the following subheading: 6.3.  Limitations and suggestions for future research. 

Author Response

Reply to Reviewer 3

Thank you for your offer. In the sixth part of the paper, I add the limitations of this study and suggestions for future research. See lines 686-696 for details, with all changes in blue font.